# Analysis of the Impact of Structural Parameter Changes on the Overall Aerodynamic Characteristics of Ducted UAVs

**Huarui Xv, Lei Zhao *, Mingjian Wu, Kun Liu, Hongyue Zhang and Zhilin Wu**

School of Mechanical Engineering, Nanjing University of Science and Technology, Nanjing 210094, China; 121101221885@njust.edu.cn (H.X.); liukun@njust.edu.cn (K.L.); wuruinan-1994@mail.njust.edu.cn (Z.W.)
* Correspondence: zhlei@njust.edu.cn

**Abstract:** Ducted UAVs have attracted much attention because the duct structure can reduce the propeller tip vortices and thus increase the effective lift area of the lower propeller. This paper investigates the effects of parameters on the aerodynamic characteristics of ducted UAVs, such as co-axial twin propeller configuration and duct structure. The aerodynamic characteristics of the UAV were analyzed using CFD methods, while the impact sensitivity analysis of the simulation data was sorted using the orthogonal test method. The results indicate that, while maintaining overall strength, increasing the propeller spacing by about 0.055 times the duct chord length can increase the lift of the upper propeller by approximately 1.3% faster. Reducing the distance between the propeller and the top surface of the duct by about 0.5 times the duct chord length can increase the lift of the lower propeller by approximately 7.7%. Increasing the chord length of the duct cross-section by about 35.3% can simultaneously make the structure of the duct and the total lift of the drone faster by approximately 150.6% and 15.7%, respectively. This research provides valuable guidance and reference for the subsequent overall design of ducted UAVs.

**Keywords:** UAVs; orthogonal experimental design; aerodynamic analysis; sliding grid





## 1. Introduction

With the advancement of modern technology and evolving human environments, UAVs have gradually become a focal point in aerospace research due to their low cost, high cost-effectiveness, and strong multi-functional cooperative capabilities. As an important category of UAVs, ducted UAVs are characterized by the fact that their propellers are enclosed within ducts [1]. This structural design provides ducted UAVs with distinct advantages compared to other types of UAVs, including higher safety, lower noise, and more compact structure [2,3]. The combination of propellers and ducted structures can significantly enhance the aerodynamic performance of UAVs. Specifically, the ducted structure can effectively suppress the propeller's tip vortex effect generated during rotation [4–6], resulting in an increased effective rotational diameter of the propeller. Additionally, the ducted structure generates additional lift under the influence of incoming airflow, which has a positive impact on vertical takeoff and landing, hovering, and maneuverability in various flight attitudes of UAVs.

The Computational Fluid Dynamics (CFD) method, as a computer-aided simulation technique, has been widely applied by researchers in the aviation field. Scholars such as Ma'arof M.I.N, Chen S.H.A., Gowtham G., and others have utilized CFD simulation to theoretically validate and elucidate the patterns of UAVs' wings, propellers, structural parameters, and other aspects, thus demonstrating that CFD is a rapid and feasible research approach [7–10]. Mishra Nirmith Kumar and other researchers have studied design standards for developing UAVs in order to achieve maximum aerodynamic efficiency. They conducted efficiency tests on NACA 4-series wings with fixed-pitch propellers and observed a thrust of 15 N at a constant operating speed of 35 km per hour [11]. Zhu et al.

studied propellers for UAVs based on the NACA 0012 symmetrical airfoil. They found that performance parameters such as thrust, drag, and power are based on blade element theory. Their research results indicate that both thrust and torque increase with increasing propeller rotational speed, RPM, and pitch coefficient [12]. Dogru et al. utilized a static pressure measurement system to evaluate the thrust of a ducted fan location on the ground effect [13,14]. Morgado et al. studied the aerodynamic efficiency of propellers for UAVs operating at higher altitudes. The wings chosen for propeller design were NACA 63A514. Computational simulation results indicated that the lift-to-drag ratio can generate a sufficient pressure difference between the upper and lower surfaces of the wing, with negligible friction effects during the process [15]. Brent et al. conducted an experimental study on fans. They used a wind tunnel to test the performance of 79 aircraft propeller blades, including tests under both dynamic and static conditions. These experiments were conducted in the low-speed wind tunnel at the University of Illinois (UIUC), with wind speeds ranging from 1500 to 7500 RPM [16]. Li Y. et al. designed a novel multi-ducted propeller structure for UAVs and compared the aerodynamics of ducted and non-ducted UAVs using simulation simulations. They further examined the optimal configuration of multiple propellers with minimal tip clearance and appropriate height difference in the duct, which significantly improved lift generation and FM efficiency, providing potential design optimization for multirotor UAVs [17]. The aforementioned work is of significant importance in understanding the overall aerodynamic characteristics of ducted UAVs. However, other researchers have primarily focused on the study of aerodynamic mechanisms of ducted UAVs and concentrated on the influence of individual factors on overall aerodynamic characteristics. In contrast, this paper places its research emphasis on analyzing multiple factors under the same conditions. Therefore, it is essential to conduct aerodynamic simulation and analysis targeting multiple structural factors, as it can provide valuable guidance for the design of future ducted UAVs.

Based on a ducted UAV designed with airfoils from the Profili airfoil database, this paper summarizes previous experiences, establishes a finite element model using simulation software, and investigates the influence of structural parameter changes on the overall aerodynamic characteristics. Additionally, the orthogonal experimental method is utilized to rank the sensitivity of each structural parameter in terms of aerodynamic characteristics. The objective is to provide valuable insights for the design of future ducted UAVs.

## 2. Theoretical Model

### 2.1. Model Establishment

This paper presents the design of a low-altitude, low-speed flying ducted UAV. Based on various assumptions and research objectives, the UAV was simplified into three components: the upper propeller, the lower propeller, and the duct structure. The simplified model of the ducted UAV used for simulation is shown in Figure 1. The propulsion structure of this ducted UAV adopted a coaxial dual-rotor configuration, where the upper and lower propellers rotate at the same speed. This configuration not only cancels out the torque but also increases the lift. Figure 2 illustrates the relative rotation direction of the coaxial dual-rotor.

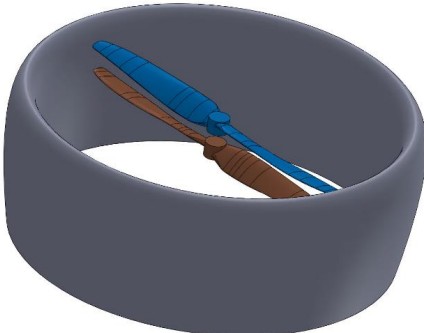

**Figure 1.** Simplified model of the ducted UAV.

Rotation direction

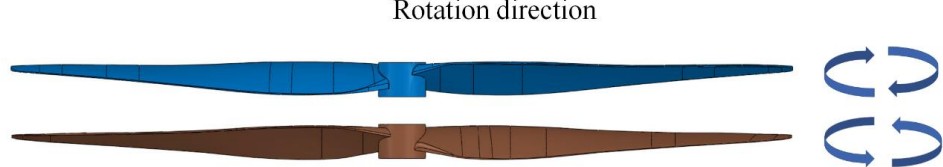

**Figure 2.** Relative rotation direction of the coaxial dual-rotor.

The UAV is powered by a DC motor, with a maximum speed of up to 9180 rpm when driven at a voltage of 12 V. The corresponding maximum tip speed of the propeller is estimated to be around 144.2 m/s. By using the Reynolds number calculation formula, it can be determined that the Reynolds number (Re) reaches approximately 206,685.3807 when the UAV flies at an altitude of 100 m.

Among them, five representative airfoil profiles were preliminarily selected from the American NACA airfoil database. These included NACA 0009 (symmetrical airfoil), NACA 0018 (symmetrical airfoil), NACA4415 (double convex airfoil), NACA6409 (concave-convex airfoil), and NACA M16 (flat-convex airfoil). The aerodynamic characteristics of these five airfoil profiles were compared under the condition of Reynolds number Re = 207,000. Figure 3 visually illustrates the variations in lift coefficient and drag coefficient for each airfoil profile. Figure 4 describes the variations in lift coefficient and drag coefficient with changing angle of attack for each airfoil profile. Figure 5 depicts the variations in lift-to-drag ratio and pitching moment coefficient with changing angle of attack for each airfoil profile.

From Figures 3–5, it can be observed that the NACA 0018 symmetrical airfoil (as shown in Figure 6) exhibited a continuous increase in the lift coefficient when increasing the angle of attack compared to other airfoil profiles. The drag coefficient showed a decreasing trend initially, followed by an increasing trend. Consequently, the lift-to-drag ratio increased initially and then decreased when increasing the angle of attack. It reached its maximum at around a 6° angle of attack, and the curve appeared relatively flat. The pitching moment, it also fluctuated around zero.

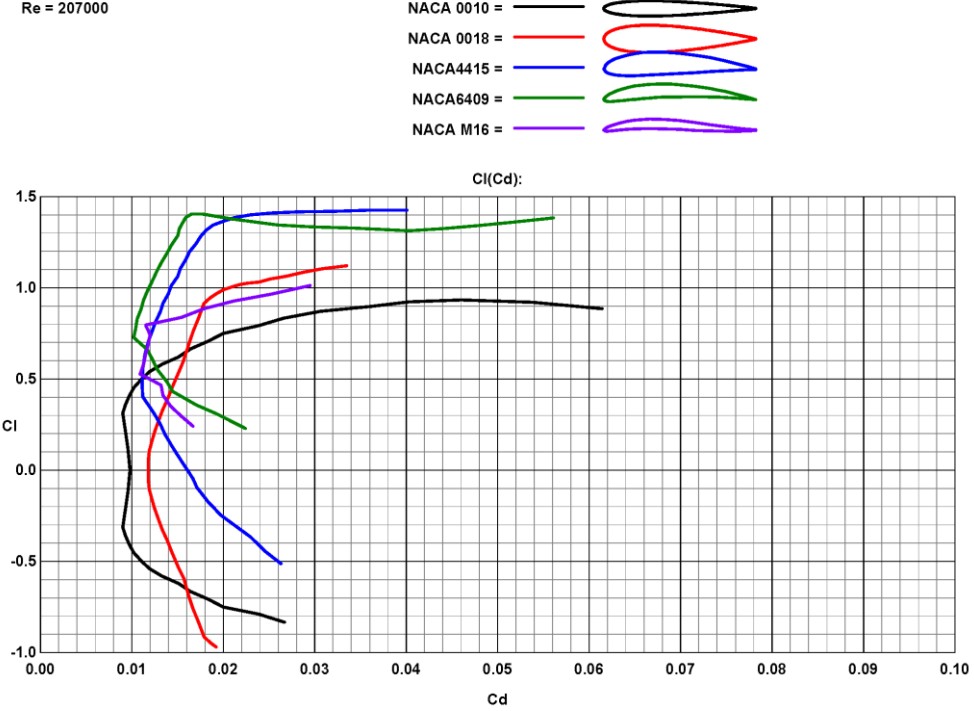

**Figure 3.** The variation relationship between the lift coefficient and drag coefficient for each airfoil profile.

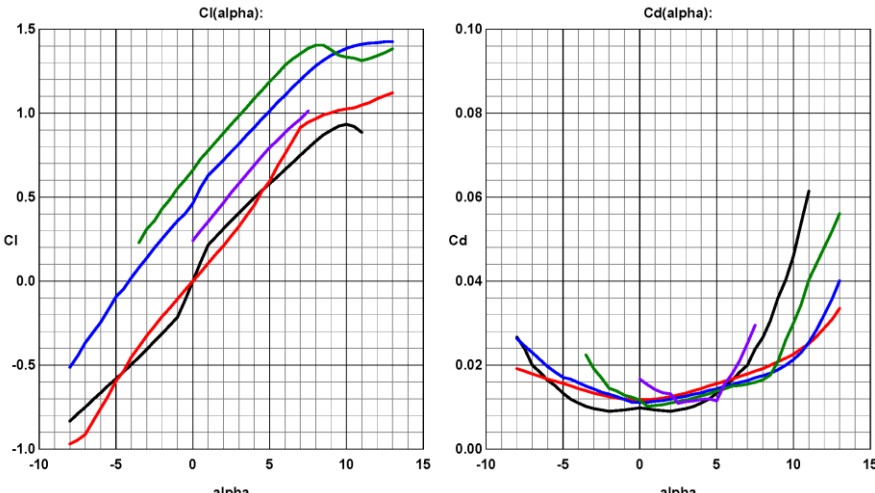

**Figure 4.** The variation pattern of the lift coefficient and drag coefficient for each airfoil profile with changing angle of attack.

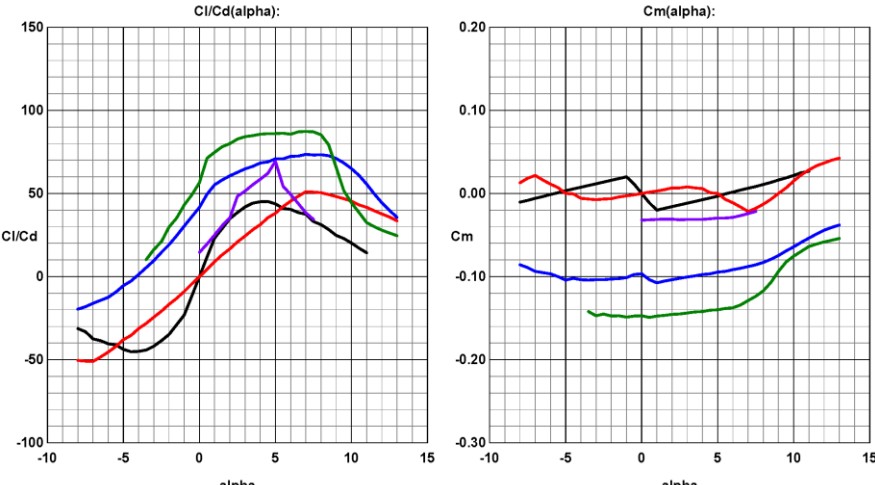

**Figure 5.** The variation pattern of lift-to-drag ratio and pitching moment coefficient for each airfoil profile with changing angle of attack.

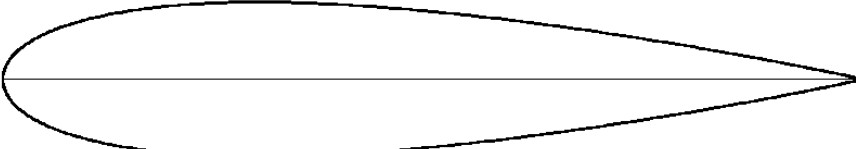

**Figure 6.** NACA 0018 airfoil.

The selected propeller model for this experiment was EP-9050. A pair of propellers is designed to form a coaxial dual-rotor structure. The propellers are made of plastic and can operate at various design voltages. Specifically, under an operating voltage of 12 V, each propeller rotated at a speed of 9180 RPM. Both propellers had a diameter of 230 mm. At a rotational speed of 9180 RPM, the designed lift of the coaxial dual-rotor was 1418 g (50.28 oz), with a design efficiency of 2.95 g/W. The relevant data for the propellers can be found in Table 1.

Due to the presence of the ducted structure, the aerodynamic characteristics of the coaxial propellers inside the duct significantly differ from those of coaxial propellers without a ducted structure. In the subsequent analysis, simulation techniques were employed to explore the impact of several structural parameters. These parameters included the

propeller spacing (D), the distance between the propeller blade tip and the duct wall (d), the distance between the propeller and the top surface of the duct (S), as well as the angle of attack (α) and chord length (L) in the duct cross-sectional configuration. Figure 7 vividly illustrates the distance positions represented by each structural parameter, and their corresponding relationships can be found in Table 2.

**Table 1.** Propeller structure and testing parameters.

| Size | Diameter (mm) | Volts (V) | Amps (A) | Watts (W) | RPM | Thrust (g) | Thrust (oz) | Efficiency (g/W) |
|---|---|---|---|---|---|---|---|---|
| GWS EP-9050 | 230 | 12.0 | 40.00 | 480 | 9180 | 1418.00 | 50.28 | 2.95 |

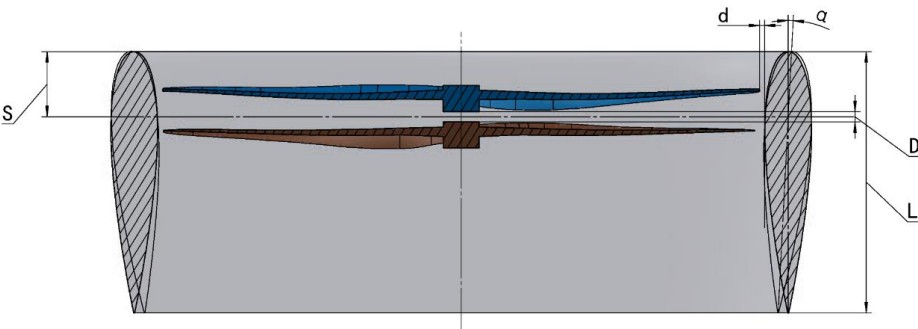

**Figure 7.** Schematic view and corresponding relationship of research parameters for the ducted UAV.

**Table 2.** Correspondence between research parameters and symbols.

| Parameters | | Symbols |
|---|---|---|
| Propeller spacing | | D |
| The distance between the propeller blade tip and the duct wall | | d |
| The distance between the propeller and the top surface of the duct | | S |
| Duct cross-sectional configuration | Angle of attack | α |
| | Chord length | L |

To facilitate analysis and calculations, the following assumptions were made for the research work on the studied ducted UAV in this paper:

1. In the overall structure, the duct and coaxial dual-rotor configuration have a relatively significant impact on the overall aerodynamic characteristics compared to other structures. Therefore, it was assumed that the influence of other structures on the overall aerodynamic characteristics could be neglected;

2. In the actual flight processes, small protrusions and indentations on components can affect the overall aerodynamic characteristics. However, these small protrusions and indentations are considered to be related to machining precision. Therefore, it was assumed that the surfaces of the duct and coaxial dual-rotor structure were smooth, free from defects, and had a high level of machining precision;

3. In practical situations, assembly errors can also cause changes in the overall aerodynamic characteristics. However, their impact is considered to be minor. Therefore, it was assumed that this UAV had a high level of assembly precision and was free from errors.

### 2.2. Control Equations

In this study, three-dimensional turbulent flow was considered, neglecting the effect of heat transfer in the air. The governing equations employed for the analysis were the Navier–Stokes (N–S) equations in fluid mechanics [18], and the Realizable *k*-*ε* turbulence model was used to solve the equations.

The theoretical formulation of the governing equations can be described as follows:
Continuity equation:

$$\frac{\partial(\rho u)}{\partial x} + \frac{\partial(\rho v)}{\partial y} + \frac{\partial(\rho w)}{\partial z} + \frac{\partial \rho}{\partial t} = 0 \tag{1}$$

Momentum conservation equation:

$$\begin{cases} \rho\left(\frac{\partial u}{\partial t} + u\frac{\partial u}{\partial x} + v\frac{\partial u}{\partial y} + w\frac{\partial u}{\partial z}\right) = \rho f_x - \frac{\partial p}{\partial x} + \mu\left(\frac{\partial^2 u}{\partial x^2} + \frac{\partial^2 u}{\partial y^2} + \frac{\partial^2 u}{\partial z^2}\right) \\ \rho\left(\frac{\partial v}{\partial t} + u\frac{\partial v}{\partial x} + v\frac{\partial v}{\partial y} + w\frac{\partial v}{\partial z}\right) = \rho f_y - \frac{\partial p}{\partial y} + \mu\left(\frac{\partial^2 v}{\partial x^2} + \frac{\partial^2 v}{\partial y^2} + \frac{\partial^2 v}{\partial z^2}\right) \\ \rho\left(\frac{\partial w}{\partial t} + u\frac{\partial w}{\partial x} + v\frac{\partial w}{\partial y} + w\frac{\partial w}{\partial z}\right) = \rho f_z - \frac{\partial p}{\partial z} + \mu\left(\frac{\partial^2 w}{\partial x^2} + \frac{\partial^2 w}{\partial y^2} + \frac{\partial^2 w}{\partial z^2}\right) \end{cases} \tag{2}$$

Energy conservation equation:

$$\begin{aligned} \frac{\partial(\rho e)}{\partial t} + \nabla\bullet(\rho e V) = {} & \rho\dot{q} - \frac{\partial}{\partial x}\left(-k\frac{\partial T}{\partial x}\right) - \frac{\partial}{\partial y}\left(-k\frac{\partial T}{\partial y}\right) - \frac{\partial}{\partial z}\left(-k\frac{\partial T}{\partial z}\right) \\ & + \tau_{xx}\frac{\partial u}{\partial x} + \tau_{yx}\frac{\partial u}{\partial y} + \tau_{zx}\frac{\partial u}{\partial z} - p_x\frac{\partial u}{\partial x} \\ & + \tau_{yy}\frac{\partial v}{\partial y} + \tau_{xy}\frac{\partial v}{\partial x} + \tau_{zy}\frac{\partial v}{\partial z} - p_y\frac{\partial v}{\partial y} \\ & + \tau_{zz}\frac{\partial w}{\partial z} + \tau_{yz}\frac{\partial w}{\partial y} + \tau_{xz}\frac{\partial w}{\partial x} - p_z\frac{\partial w}{\partial z} \end{aligned} \tag{3}$$

*2.3. Turbulence Model*

The *k-ε* turbulence model, widely used as a two-equation model, is a semi-empirical formulation based on turbulent kinetic energy (*k*) and turbulence dissipation rate (*ε*). The equation for *k* is derived from exact equations, while the equation for *ε* is derived from empirical formulas. The Realizable *k*-epsilon model used in this study incorporates the effects of mean rotation in the definition of turbulent viscosity in the standard k-epsilon model [19].

The theoretical formulation of the turbulence model can be described as follows:
*k*-equation:

$$\frac{\partial(\rho K)}{\partial t} + \frac{\partial\left(\rho\overline{u_j}K\right)}{\partial x_j} = \frac{\partial}{\partial x_j}\left[\left(\mu + \frac{\mu_t}{Pr_K}\right)\frac{\partial K}{\partial x_j}\right] + P_K + G_b - \rho\varepsilon - Y_M \tag{4}$$

*ε*-equation:

$$\begin{aligned} \frac{\partial(\rho\varepsilon)}{\partial t} + \frac{\partial\left(\rho\overline{u_j}\varepsilon\right)}{\partial x_j} = {} & \frac{\partial}{\partial x_j}\left[\left(\mu + \frac{\mu_t}{Pr_\varepsilon}\right)\frac{\partial\varepsilon}{\partial x_j}\right] + \rho C_1\overline{S}\varepsilon - C_2\rho\frac{\varepsilon^2}{K+\sqrt{v\varepsilon}} \\ & + C_{\varepsilon 1}\frac{\varepsilon}{k}C_{\varepsilon 3}G_b \end{aligned} \tag{5}$$

Among them: $C_1 = \max\left[0.43, \frac{\eta}{\eta+5}\right]$, $\eta = S\frac{k}{\varepsilon}$; $S = \sqrt{2S_{ij}S_{ij}}$, $\mu_t = C_\mu\rho\frac{K^2}{\varepsilon}$, $C_\mu = \frac{1}{A_0 + A_S\frac{U^*K}{\varepsilon}}$, $U^* = \sqrt{S_{ij}S_{ij} + \widetilde{\Omega_{ij}}\widetilde{\Omega_{ij}}}$, $\widetilde{\Omega_{ij}} = \Omega_{ij} - 2\varepsilon_{ijk}\omega_k$, $\Omega_{ij} = \overline{\Omega_{ij}} - \varepsilon_{ijk}\omega_k$, $A_0 = 4.04$, $A_S = \sqrt{6}\cos\varphi$, $\varphi = \frac{1}{3}\arccos(\sqrt{6}W)$, $W = \frac{S_{ij}S_{jk}S_{ki}}{\widetilde{S}^3}$, $\widetilde{S} = \sqrt{S_{ij}S_{ij}}$, $S_{ij} = \frac{1}{2}\left(\frac{\partial u_j}{\partial x_i} + \frac{\partial u_i}{\partial x_j}\right)$; $C_{1\varepsilon} = 1.44$, $C_2 = 1.9$, $\sigma_k = 1.0$, $\sigma_\varepsilon = 1.2$.

Compared to the standard *k*-epsilon model, the main variations of the Realizable *k*-epsilon model are as follows:

1. The calculation formula for turbulent kinematic viscosity undergoes changes, introducing variables related to rotation;
2. The epsilon equation undergoes significant changes, with the production term no longer including the generation term from the *k* equation. The new form of the equation can better handle the information transformation at the sliding mesh boundary;
3. The second-to-last term in the epsilon equation does not exhibit any singularity. Even when *K* is very small or zero, the denominator will not be zero.

Based on the aforementioned major changes, this model can more accurately predict turbulence behavior in complex flow conditions and provide more reliable numerical results. Whether dealing with turbomachinery, aerospace, or other fields related to these flows, the Realizable *k*-epsilon model has been widely applied and achieved good results.

*2.4. Boundary Conditions*

The simulation involved six types of boundary conditions, including velocity inlet, pressure outlet, wall, interface, rotational speed, and far-field boundary conditions. The outer flow field and the two inner flow fields were all cylindrical in shape. The upper surface of the outer flow field was set as the velocity inlet, while the lower surface was set as the pressure outlet. The surface of the cylinder was defined as the wall, and the remaining parts, where the flow fields interact, were set as interfaces. At the velocity inlet, the inflow velocity was specified, and the reference pressure was set to 0. Similarly, the static pressure at the pressure outlet was also set to 0. The wall was defined as a non-penetrable, viscous, no-slip surface. The rotational speed was set to the actual speed of the propeller. The far-field boundary condition was specified as a free surface, allowing fluid to pass through freely.

## 3. Numerical Simulation

### 3.1. Grid Partitioning

Based on the structure of the propeller and duct, a finite element model for dynamic analysis was established using Ansys Fluent 2022 R1.

As shown in Figure 8, the coaxial dual-rotor structure without a duct adopted an edge-refined, unstructured polyhedral grid. It can be observed that the grid became denser near the edge region for accurate calculations. The entire flow field domain consisted of five parts, with two inner flow field domains (upper inner flow field domain, lower inner flow field domain), each enveloping a wall structure (upper propeller, lower propeller). The outer flow field domain encompassed the inner flow field domains and all wall structures. The inner flow field domains enclosing the upper and lower propellers were rotating domains and were set to different speeds in subsequent simulation validation and experiments. The information exchange between the internal flow field domains was carried out through intersecting planes (upper inner flow field domain lower surface and lower inner flow field domain upper surface), and the internal flow field domains were enclosed by the external flow field and shared information through another pair of intersecting planes. In the simulation solution, the number of grids was approximately 1.15 million.

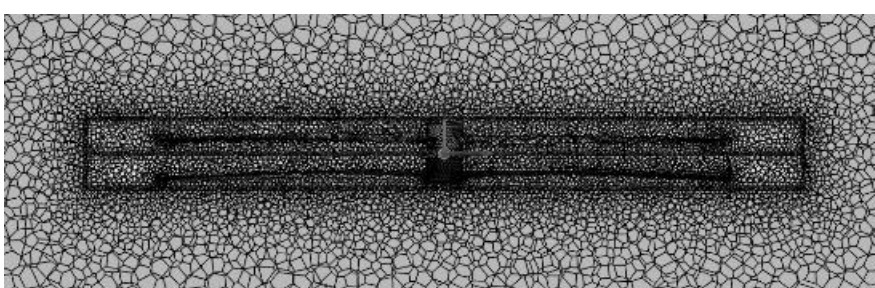

**Figure 8.** The flow field grid for the coaxial dual-rotor structure without a duct.

Similarly, as shown in Figure 9, the ducted UAV, compared to the aforementioned coaxial dual-rotor structure, included a duct structure and still adopted an edge-refined, unstructured polyhedral grid. The entire flow field domain consisted of six parts, with three flow field domains (upper inner flow field domain, lower inner flow field domain, outer flow field domain), each enveloping a wall structure (upper propeller, lower propeller, entire flow field, and wall structures). The flow field domains enclosing the upper and lower propellers were rotating domains, and the internal flow field domains exchanged information through intersecting planes (upper inner flow field domain lower surface

and lower inner flow field domain upper surface). The internal flow field domains were enclosed by the external flow field and shared information through intersecting planes in the upper and lower regions. In the simulation solution, the average number of grids was approximately 3 million.

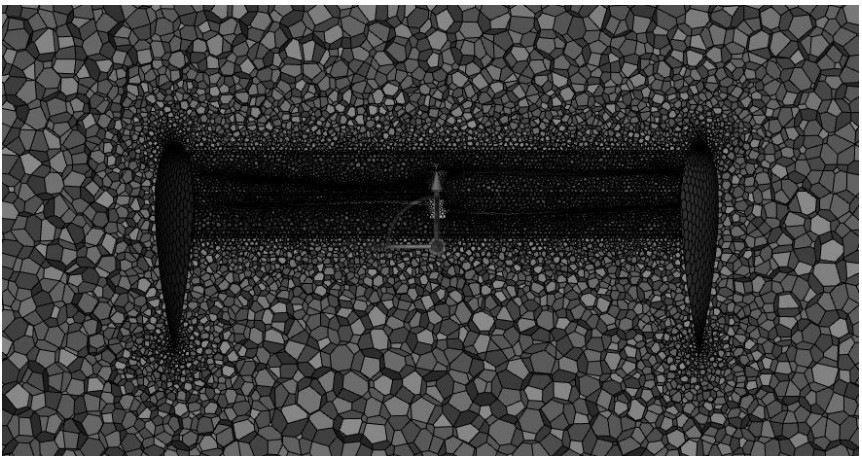

**Figure 9.** The flow field grid for the ducted coaxial dual-rotor configuration.

In addition, to reduce potential errors, the simulation results were obtained by averaging multiple node calculations.

### 3.2. Grid Independence and Time Sensitivity Testing

Considering the issues of grid independence and sensitivity of simulation results to time steps, validation tests were conducted for the simulation of unsteady aerodynamic characteristics of the coaxial propellers. In the testing process, three different grid systems were employed with resolutions of 360,000 (365,558) cells, 1.15 million (1,151,946) cells, and 6.86 million (6,869,742) cells, respectively. The entire flow field could be divided into three main domains, including two internal flow domains enveloping the coaxial propellers and an outer flow domain enveloping the external surfaces. The partitioning of these three domains generally followed the ratio of the total number of grids. Each grid block was interconnected using sliding mesh technology to calculate the aerodynamic state for each time step. Additionally, three different time steps were tested with a grid size of 1.15 million (1,151,946) cells. The time steps used were 0.0005 s, 0.001 s, and 0.002 s, respectively.

Figures 10 and 11 depict the lift comparison of the upper and lower blades of a coaxial propeller under different grid densities and time steps when the flow reached a stable state. In this simulation, both the grid independence and time step sensitivity tests exhibited similar simulation results once stabilized. However, the configuration with a grid density of 1.15 million (1,151,946) cells and a time step of 0.002 s demonstrated higher stability and better data convergence. Based on simulation experience analysis and stability considerations, the partitioning scheme with 1.15 million (1,151,946) cells and a time step of 0.002 s was chosen as the optimal simulation setup. The simulation process, occupying 30 cores for computation, took an average of 5 h to complete, ensuring both numerical accuracy and experimental feasibility in terms of space and time.

### 3.3. Simulation Validation

To verify the accuracy of the finite element model, a comparative experimental test, as shown in Figure 12, was designed in this study. Using the same workstation, simulations were conducted for all operating conditions. Three sets of experiments were performed using a torque motor and a high-frequency testing platform to measure the lift of a single upper rotor, a single lower rotor, and coaxial dual rotors at 2000 rpm, 4000 rpm,

6000 rpm, 8000 rpm, and 9180 rpm. The hardware configuration parameters/indicators of the workstation can be found in Table 3.

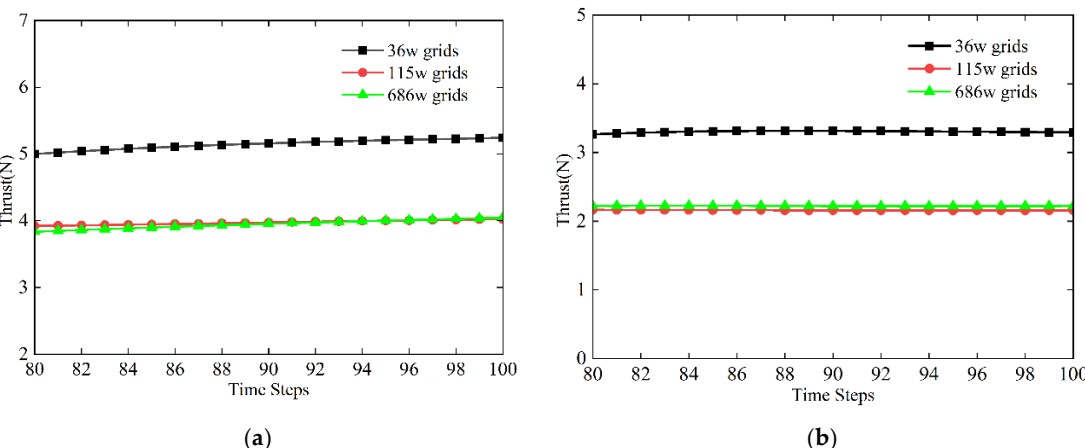

(**a**)　　　　　　　　　　　　　　(**b**)

**Figure 10.** The influence of grid density on the lift of coaxial propellers: (**a**) Influence diagram of the upper propeller; (**b**) Influence diagram of the lower propeller.

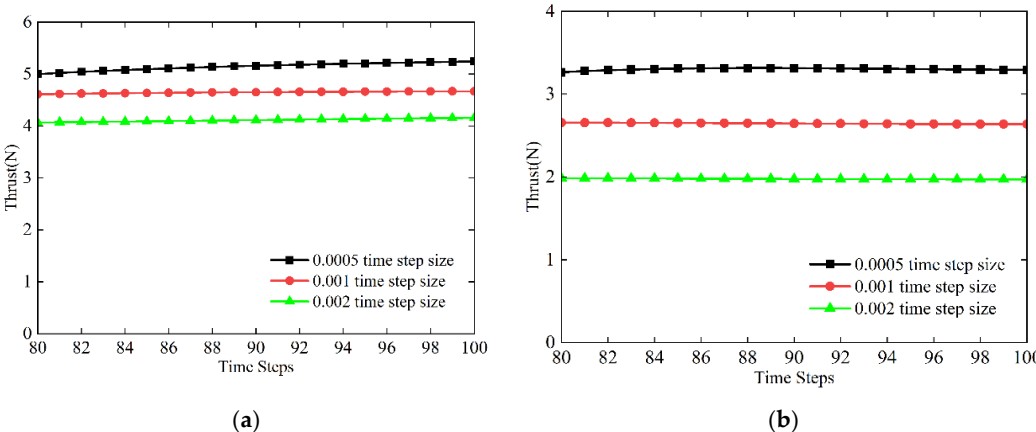

(**a**)　　　　　　　　　　　　　　(**b**)

**Figure 11.** The influence of time step variation on the lift of coaxial propellers: (**a**) Influence diagram of the upper propeller; (**b**) Influence diagram of the lower propeller.

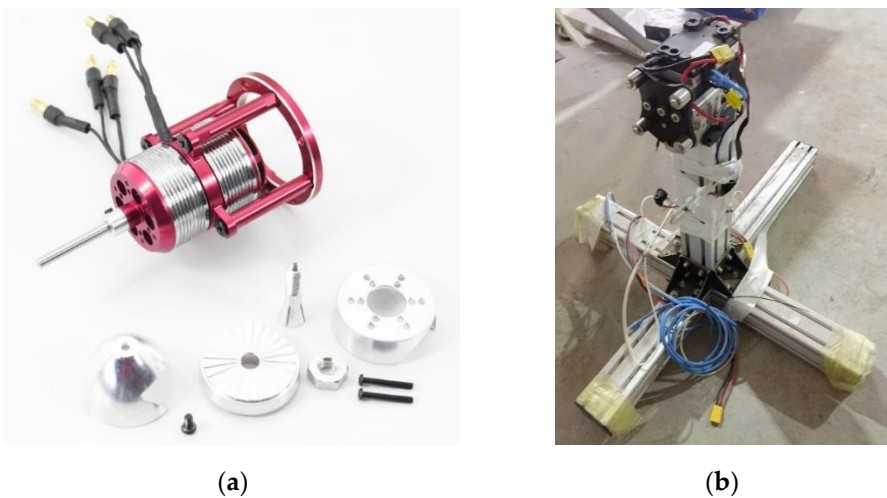

(**a**)　　　　　　　　　　　　　　(**b**)

**Figure 12.** Experimental equipment: (**a**) Torque motor; (**b**) High-frequency testing platform.

**Table 3.** Hardware configuration parameters/indicators of the workstation.

| Hardware Configuration | Parameter Indicators |
| --- | --- |
| CPU | Intel(R) Xeon(R) Gold 6240R @ 2.40 GHz |
| Operating system | Windows 10 Professional |
| RAM capacity | 384 GB |

Figure 13 illustrates the lift variation curves of the individual propeller and the coaxial dual-rotor without a duct between speeds of 2000 rpm and 8000 rpm. It can be observed that there was good agreement between the experimental and simulation results, with the maximum error occurring at the speed of 9180 rpm, which did not exceed 2%. This indicated that the simulation model in this study can provide reasonably accurate aerodynamic characteristics. As all simulation results aligned well with the experimental results, it demonstrates the effectiveness of the numerical simulation method employed in this study. The discrepancies between the experimental and simulation results can be attributed to errors in the turbulence model, assumptions made in the simulation, and variations in the experimental environment.

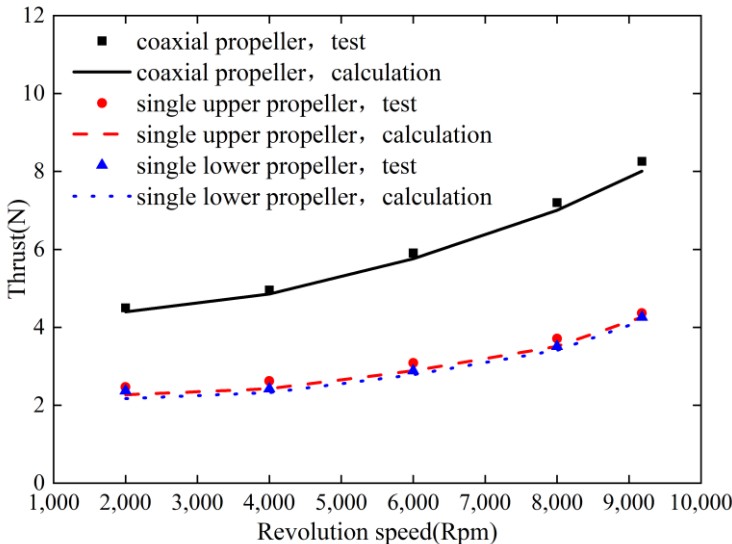

**Figure 13.** Comparison of lift between single propeller and coaxial dual-rotor.

## 4. The Aerodynamic Characteristics of a Ducted UAV

### 4.1. Brief Description of Aerodynamic State in Coaxial Dual-Rotor Configuration within Duct

Figure 14 illustrates the slipstream model of the coaxial dual-rotor configuration in a duct. The upper rotor is primarily affected by the slipstream generated by the upper airflow along the duct, the side inflow of the lateral air, and the tip vortices generated by the slipstream itself and the disturbance from the lower rotor. On the other hand, the lower rotor is mainly influenced by the inflow from the upper rotor and its own tip vortices. Lift is generated by the interaction among the upper rotor, lower rotor, and the duct structure in a complex flow environment.

### 4.2. Analysis of Aerodynamic Advantages of Ducted UAVs

In this section, a comparative analysis of the aerodynamic characteristics of ducted UAVs is conducted based on simulation results. This analysis aims to identify the advantages of ducted UAVs compared to the structure of coaxial dual-rotor UAVs without a duct. Furthermore, the impact of changes in structural parameters of ducted UAVs on overall aerodynamic characteristics and the ranking of factors' sensitivity are systematically discussed.

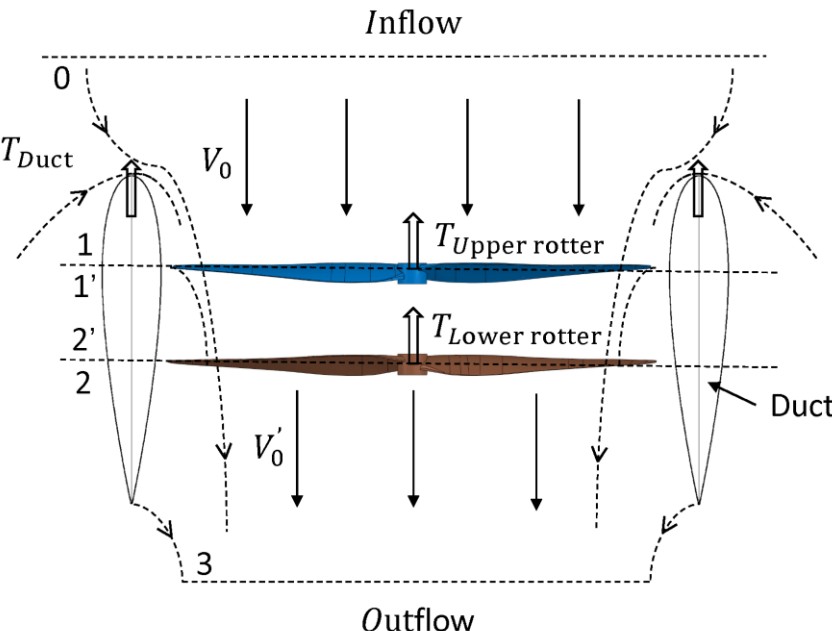

**Figure 14.** Slipstream model of the coaxial dual-rotor configuration in a duct.

Figures 15 and 16 depict the computed results of static pressure contours for the single propeller without a duct and the coaxial dual-rotor without a duct at a speed of 9180 rpm. The pressure on the upper and lower blades of the coaxial dual-rotor showed an increasing trend due to their mutual interaction. The pressure on the upper surface of both propellers increased, and under the influence of the upper propeller, the pressure on the upper surface of the lower propeller increased more rapidly, particularly evident in the outer wing region, especially near the leading edge. This phenomenon is associated with the effective lifting surface, as the airflow conditions for the lower propeller are more complex, resulting in higher pressure. The results indicated that the total lift of the coaxial dual-rotor without a duct increased by 87.95% and 92.47% compared to the lift of the single propeller without a duct (upper and lower, respectively). However, due to the mutual interaction, the lift of the coaxial upper and lower propellers without a duct was reduced by 5.10% and 5.23%, respectively, compared to the lift of the single propeller without a duct (upper and lower).

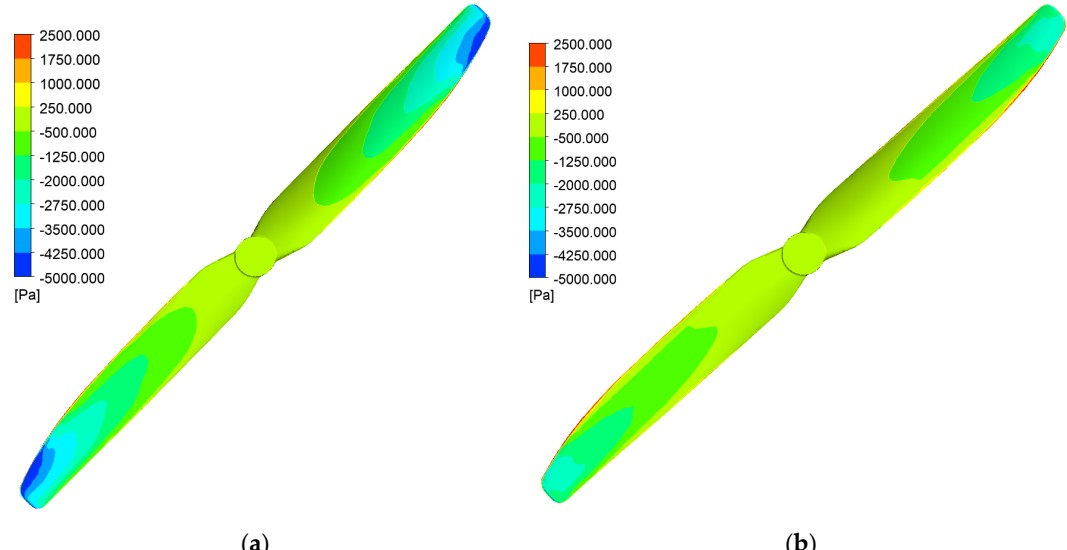

(**a**)　　　　　　　　　　　　　　　　(**b**)

**Figure 15.** Pressure distribution contour of the upper propeller: (**a**) Influence diagram of the single upper propeller; (**b**) Influence diagram of the coaxial dual-rotor upper propeller.

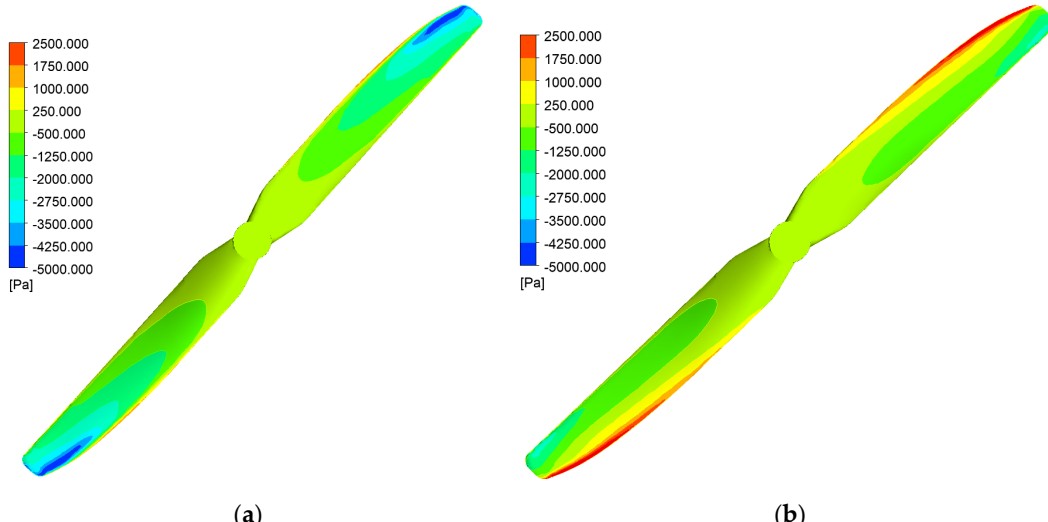

(**a**)                                                                    (**b**)

**Figure 16.** Pressure distribution contour of the lower propeller: (**a**) Influence diagram of the single lower propeller; (**b**) Influence diagram of the coaxial dual-rotor lower propeller.

Figures 17 and 18 illustrate the computed results of static pressure contours for the coaxial dual-rotor without a duct and the coaxial dual-rotor with a duct at a speed of 9180 rpm. The pressure on the upper and lower blades of the coaxial dual-rotor with a duct showed little difference under mutual interaction but still exhibited a decreasing trend. This is because the duct structure effectively reduces the impact of the upper propeller's tip vortex, thereby reducing the pressure on the lower propeller. The results indicated that the total lift of the coaxial dual-rotor with a duct increased by 25.46% compared to the total lift of the coaxial dual-rotor without a duct. Due to the influence of the duct structure, the lift of the coaxial upper and lower propellers with a duct increased by 3.08% and 2.41%, respectively, compared to the lift of the coaxial upper and lower propellers without a duct.

Figures 19 and 20 display the pressure contour and velocity vector maps of the coaxial dual-rotor without a duct and the coaxial dual-rotor with a duct. It can be observed that the tip vortex of the lower propeller without a duct was significantly larger than that of the upper propeller, which is one of the reasons for the downstream airflow contraction beneath the upper propeller. However, due to the presence of the duct structure, the magnitude of the tip vortex vorticity was reduced by 76.3%, and the effective lifting surface of the lower propeller extended along its entire length.

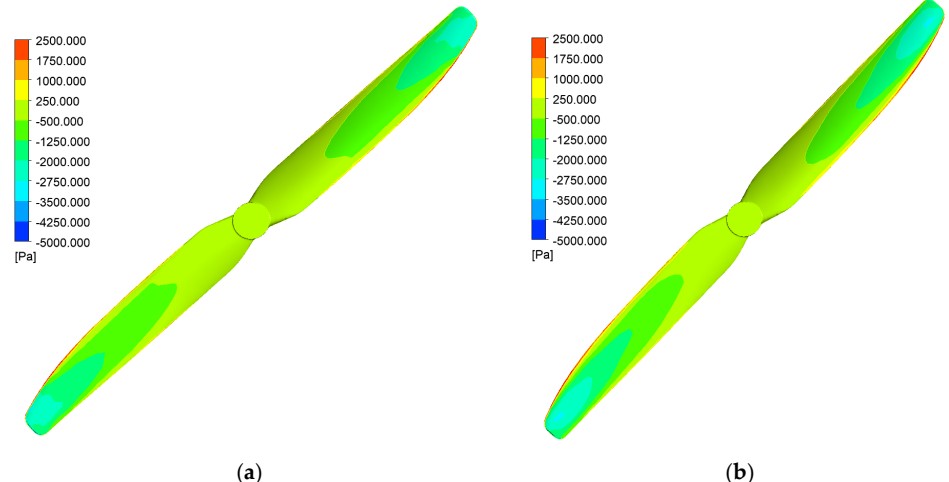

(**a**)                                                                    (**b**)

**Figure 17.** Pressure distribution contour of the upper propeller: (**a**) Influence diagram of the upper propeller without a duct; (**b**) Influence diagram of the upper propeller with a duct.

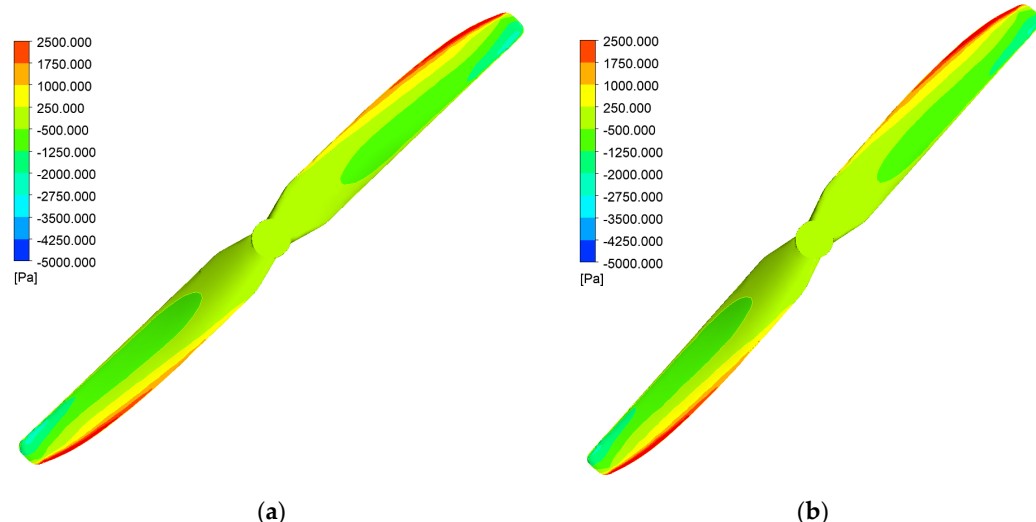

(**a**)  (**b**)

**Figure 18.** Pressure distribution contour of the lower propeller: (**a**) Influence diagram of the lower propeller without a duct; (**b**) Influence diagram of the lower propeller with a duct.

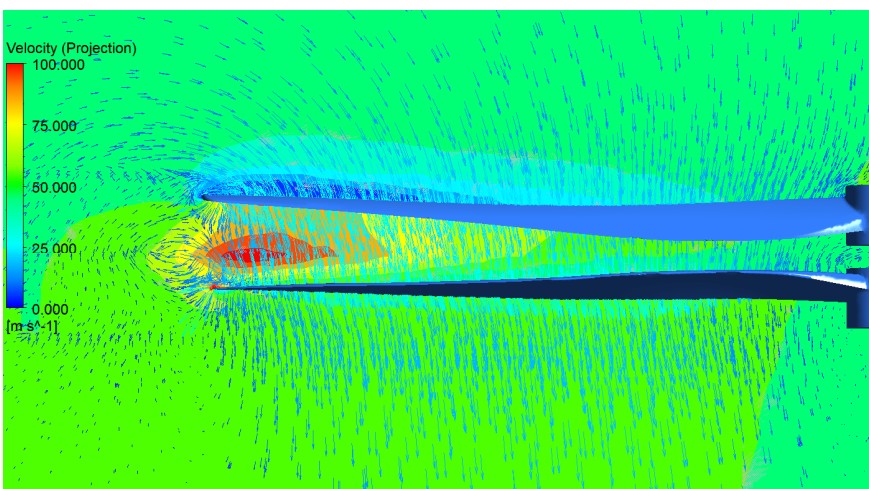

**Figure 19.** Pressure contour and velocity vector maps of the coaxial dual-rotor without a duct.

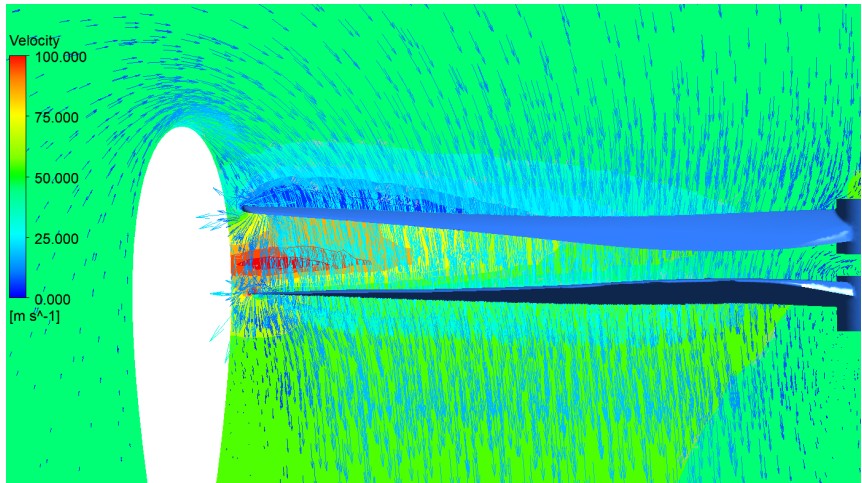

**Figure 20.** Pressure contour and velocity vector maps of the coaxial dual-rotor with a duct.

Figure 21 illustrates the vorticity contour of the coaxial dual-rotor without a duct and the coaxial dual-rotor with a duct. The results showed that the vorticity magnitude at the trailing edge of the coaxial dual-rotor without a duct was 5.2% greater than that of the coaxial dual-rotor with a duct. Compared to the coaxial dual-rotor with a duct, the mutual interaction between the upper and lower blades of the coaxial dual-rotor without a duct was more significant, and there was a higher level of vorticity dissipation. In contrast, the coaxial dual-rotor with a duct exhibited more concentrated vorticity.

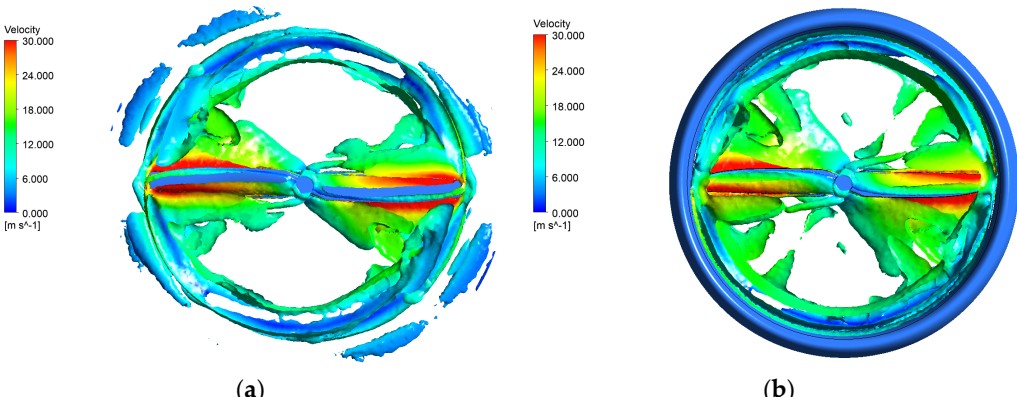

(**a**)  (**b**)

**Figure 21.** Contour maps of vorticity for coaxial dual-rotor: (**a**) Vorticity map without a duct structure; (**b**) Vorticity map with a duct structure.

### 4.3. The Impact of Multi-Factor Changes on Aerodynamics

In order to investigate the impact of changes in rotor spacing on overall aerodynamics, 12 sets of simulation scenarios were established. The standard value was set at 4 mm, and variations of ±0.5 mm were applied incrementally. Figure 22 illustrates the trend of the simulation results. Specifically, within the range of 2.5% to 8% of the chord length distance, both the upper and lower propellers exhibited increasing lift forces. However, the increase in lift force for the upper propeller was relatively slow, while the lower propeller showed a faster increasing trend. The lift force of the ducted structure remained relatively stable, and changes in rotor spacing had minimal impact on it. The total lift force continuously increased under the combined influence of these three lift forces.

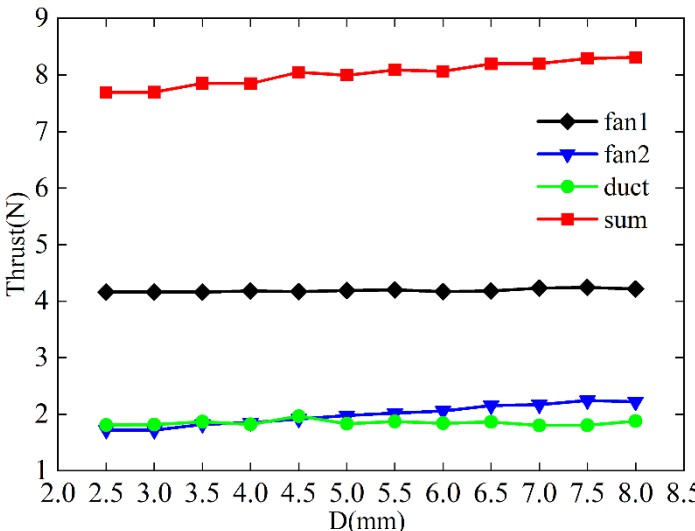

**Figure 22.** The influence of rotor spacing on the lift forces of each research object.

In order to investigate the impact of changes in the distance between the rotor blade tip and the duct wall on overall aerodynamics, 7 sets of simulation scenarios were established.

The standard value was set at 2 mm, and variations of $\pm 0.5$ mm were applied incrementally. Figure 23 illustrates the trend of the simulation results. Specifically, within the range of 0.5% to 3.5% of the chord length distance, both the upper and lower propellers exhibited a slow and uneven increase in lift force. However, the lift force of the ducted structure continuously decreased as the distance increased. The total lift force decreased under the combined influence of these three lift forces.

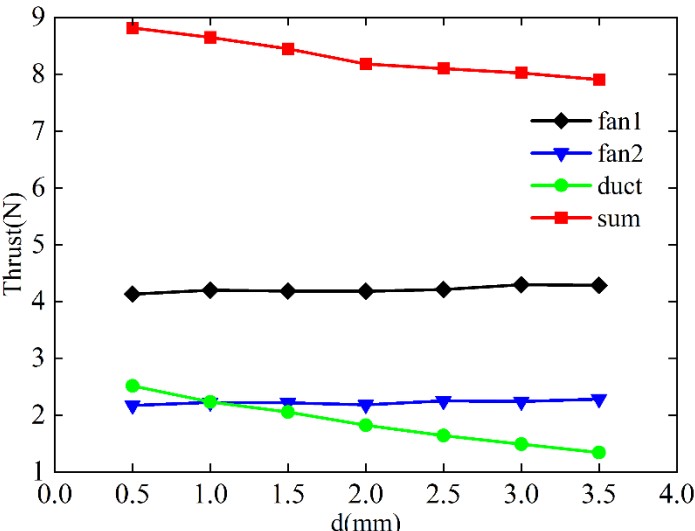

**Figure 23.** The influence of the distance between the rotor blade tip and duct wall on the lift forces of each research object.

In order to investigate the impact of changes in the distance between the rotor and the duct upper surface on overall aerodynamics, 7 sets of simulation scenarios were established. The standard value was set at 1/4 of the duct chord length distance, and variations of $\pm 1/12$ of the duct chord length distance were applied incrementally. Figure 24 illustrates the trend of the simulation results. Specifically, within the range of 1/12 to 7/12 of the duct chord length distance, the lift force of the upper propeller decreased continuously. The lift force of the lower propeller initially decreased and then increased at a slower rate. The lift force of the ducted structure showed significant changes, initially increasing and then decreasing as the distance increased. The total lift force initially increased and then decreased under the combined influence of these three lift forces.

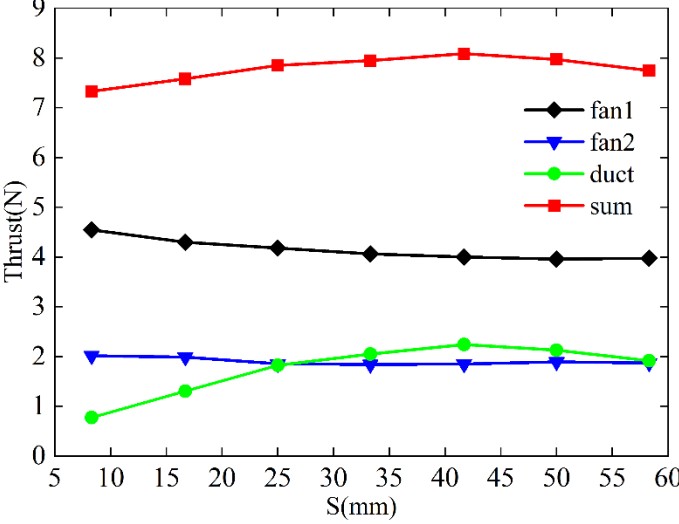

**Figure 24.** The influence of distance factors between propellers and the upper surface of the duct.

In order to investigate the influence of changes in the attack angle factor on the overall aerodynamics of the ducted section, 7 sets of simulation scenarios were established. The standard value was set at 0°, and increments and decrements of 1° were chosen for the left and right sides. Figure 25 illustrates the variation trend of the simulation results. Specifically, as the attack angle of the ducted section increased from −3° to 3°, the lift of the upper propeller increased first and then decreased gradually. The lift of the lower propeller continuously decreased. The variation in lift of the duct structure was relatively prominent, increasing consistently with the increase in attack angle. The total lift increased continuously under the combined influence of the three structural lifts.

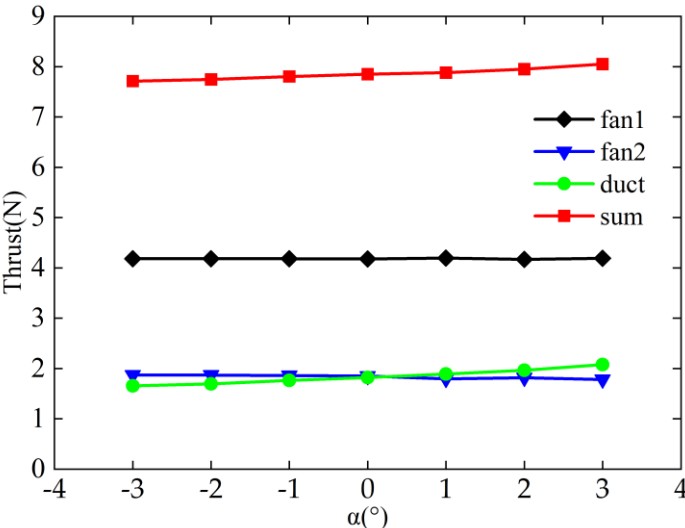

**Figure 25.** The influence of the attack angle factor of the duct section on the lift of various research objects.

In order to investigate the influence of changes in the chord length factor of the duct section on overall aerodynamics, 7 sets of simulation scenarios were established. The standard value was set at 100 mm, and increments and decrements of 5 mm were chosen for the left and right sides. Figure 26 depicts the variation trend of the simulation results. Specifically, as the chord length of the duct section increased from 85 mm to 115 mm, the lift of both the upper and lower propellers gradually decreased at a slow rate. The variation in lift of the duct structure was relatively prominent, increasing consistently with the increase in chord length. The total lift increased continuously under the combined influence of the three structural lifts.

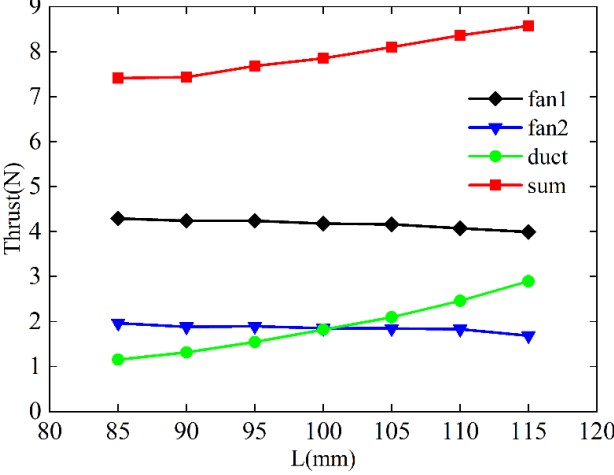

**Figure 26.** The influence of the chord length factor of the duct section on the lift of various research objects.

## 5. Orthogonal Experimental Design

Building on Section 4, in order to systematically analyze the influence of various structural parameters on the lift of ducted UAVs and obtain the relative sensitivity order of parameter changes with respect to each structural parameter, an orthogonal experimental design was employed for further investigation.

### 5.1. Factor, Level, and Index Settings

Due to the sensitivity analysis of the parameters in Section 4, the parameters studied in Section 4 were treated as factors in Section 5. As shown in Table 4, the levels of each factor were primarily based on the parameter values displayed in the Level 3 column as reference values, with step-wise extensions to the left and right.

**Table 4.** Orthogonal experimental method with level design.

| Factors | | Symbols | Levels | | | | |
|---|---|---|---|---|---|---|---|
| | | | 1 | 2 | 3 | 4 | 5 |
| Propeller spacing | | D | 3 | 3.5 | 4 | 4.5 | 5 |
| The distance between the propeller blade tip and the duct wall | | d | 1 | 1.2 | 2 | 2.5 | 3 |
| The distance between the propeller and the top surface of the duct | | S | $\frac{1}{6}L$ | $\frac{1}{4}L$ | $\frac{1}{3}L$ | $\frac{1}{2}L$ | $\frac{7}{12}L$ |
| Duct cross-sectional configuration | Angle of attack | $\alpha$ | $-2$ | $-1$ | 0 | 1 | 2 |
| | Chord length | L | 90 | 95 | 100 | 105 | 110 |

The research index is the lift of the ducted UAV, which includes the lift of the upper propeller, the lift of the lower propeller, the lift of the duct structure, and the total lift.

### 5.2. Orthogonal Table

In the orthogonal table $L_n(m^k)$, *n* represents the number of experiments, which corresponds to the number of orthogonal level combinations. *m* denotes the number of levels for each factor, while *k* indicates the maximum number of factors that can be analyzed using the table. In this case, we have 5 factors, and each factor has 5 levels [18–20]. Therefore, we needed to conduct an orthogonal experimental analysis based on the $L_{25}(5^6)$ orthogonal table. The specific operating conditions can be found in Appendix A, Table A1, and the combinations of various simulation research indices are listed on the right side.

### 5.3. Sensitivity Analysis of Factors

The sensitivity analysis of factors in orthogonal experimental design can be approached in four main ways: range analysis; standard deviation analysis; normalized range analysis; and normalized standard deviation analysis [20–22]. However, in this research problem, the presence of the numerical value 0 ($\alpha = 0°$) for the standard value made normalization unfeasible. Both range analysis and standard deviation analysis are scientifically valid methods, with the difference lying in the statistical concepts utilized. Standard deviation, as compared to range, better reflects the dispersion of a dataset in statistics. Therefore, in this section, the primary analysis was carried out using the standard deviation analysis method, while range analysis was employed for verification purposes.

$$R_j = \max(\overline{K}_{1j}, \overline{K}_{2j}, \overline{K}_{3j}, \overline{K}_{4j}, \overline{K}_{5j}) - \min(\overline{K}_{1j}, \overline{K}_{2j}, \overline{K}_{3j}, \overline{K}_{4j}, \overline{K}_{5j}) \tag{6}$$

$$\sigma_j = \sqrt{\frac{\sum\limits_{k=1}^{5} (\overline{K}_{ij} - AVE_j)^2}{4}} \tag{7}$$

From the computed results, it can be observed that both of these analysis methods effectively assessed the influence of each factor on the research object and provided similar

conclusions in terms of ranking. This consistency reinforces the reliability of sensitivity rankings for factors and provides more trustworthy decision-making criteria.

The analysis and calculation results for the lift of the upper propeller can be found in Appendix A, Table A2. Figure 27 visually illustrates the differences in range and standard deviation among factors for the research index of the upper propeller lift. The results obtained from range analysis and standard deviation analysis revealed that the sensitivity order of the factors was as follows: the spacing between propellers had the highest sensitivity; followed by the chord length of the duct section; and the distance between the propeller tip and the duct wall had relatively lower sensitivity.

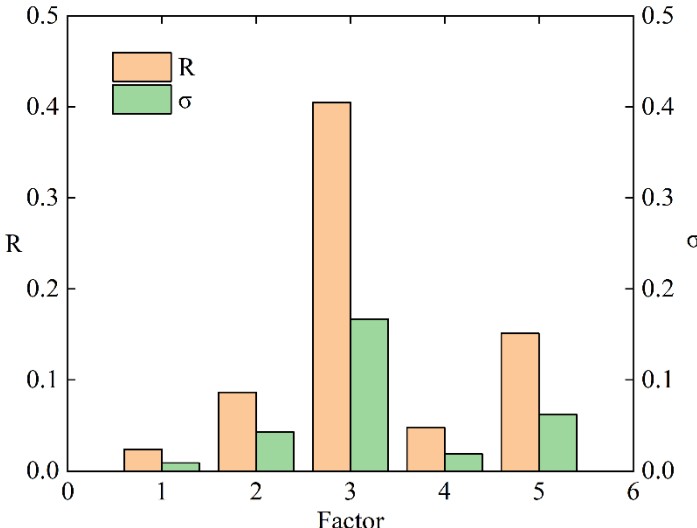

**Figure 27.** Sensitivity analysis of the influence of five research factors on the lift of the upper propeller.

The analysis and calculation results for the lift of the lower propeller can be found in Appendix A, Table A3. Figure 28 visually illustrates the differences in range and standard deviation among factors for the research index of the lower propeller lift. The results obtained from range analysis and standard deviation analysis revealed that the sensitivity order of the factors was as follows: the distance between the propeller and the top surface of the duct had the highest sensitivity; followed by the chord length of the duct section; and the distance between the propeller tip and the duct wall had relatively lower sensitivity.

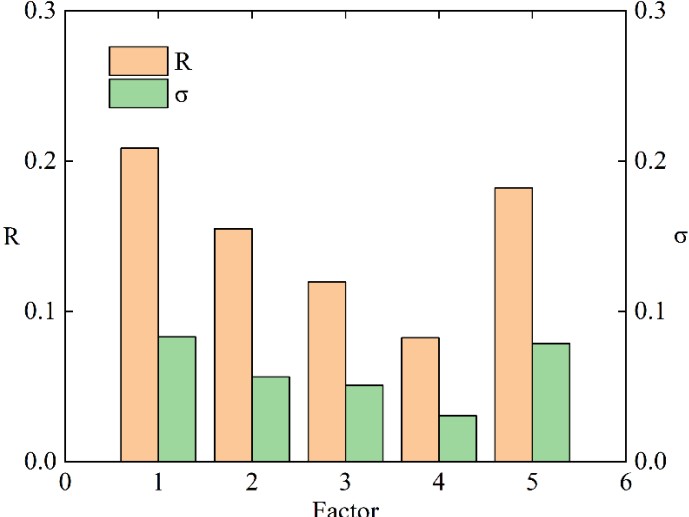

**Figure 28.** Sensitivity analysis of the influence of five research factors on the lift of the lower propeller.

The analysis and calculation results for the lift of the duct structure can be found in Appendix A, Table A4. Figure 29 visually illustrates the differences in range and standard deviation among factors for the research index of the duct structure lift. The results obtained from range analysis and standard deviation analysis revealed that the sensitivity order of the factors was as follows: the chord length of the duct section had the highest sensitivity; followed by the distance between the propeller and the top surface of the duct; and the distance between the propeller tip and the duct wall has relatively lower sensitivity.

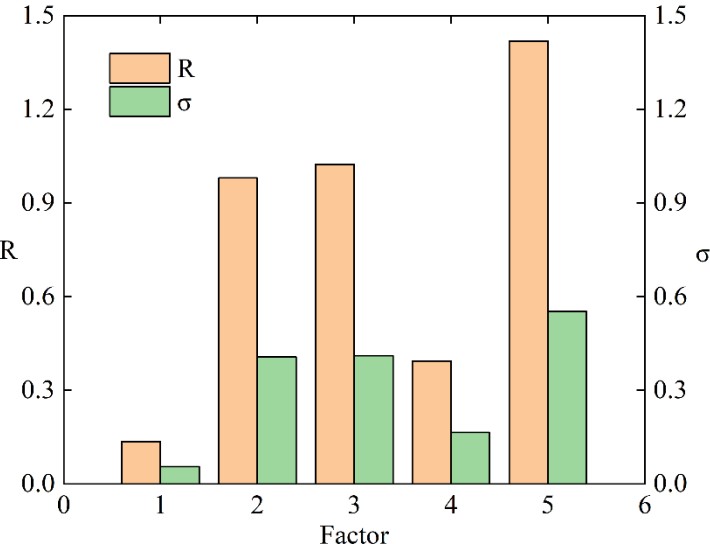

**Figure 29.** Sensitivity analysis of the influence of five research factors on the lift of the duct structure.

The analysis and calculation results for the total lift can be found in Appendix A, Table A5. Figure 30 visually illustrates the differences in range and standard deviation among factors for the research index of the total lift. The results obtained from range analysis and standard deviation analysis revealed that the sensitivity order of the factors was as follows: the chord length of the duct section had the highest sensitivity; followed by the distance between the propeller tip and the duct wall; and the distance between the propeller and the top surface of the duct had relatively lower sensitivity.

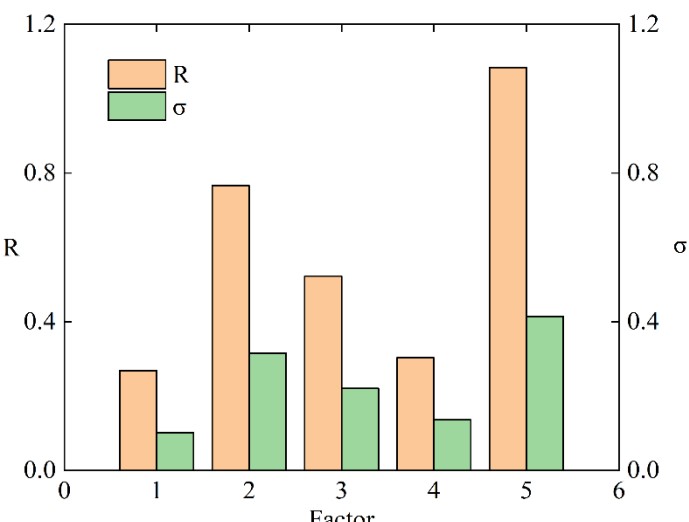

**Figure 30.** Sensitivity analysis of the influence of five research factors on the total lift of the UAV.

Based on the conclusions, structural optimization can be performed. Please refer to Table 5 for specific details.

**Table 5.** Hardware configuration parameters/indicators of the workstation.

| Parameters | D | d | S | $\alpha$ | L | Tfan1 | Tfan2 | Tduct | Tall |
|---|---|---|---|---|---|---|---|---|---|
| Numeric | 4.5 | 1 | 55 | −1 | 110 | 3.8750 | 1.7062 | 3.6990 | 9.2802 |

## 6. Conclusions

This paper conducted experimental verification and finite element modeling of ducted drones. Approximately 80 operating conditions were designed for simulation data analysis and sensitivity ranking. A novel approach combining orthogonal experimental design with the investigation of aerodynamic characteristics of ducted drones was implemented. For the first time, multiple parameters were systematically explored under the same conditions. The main conclusions are as follows:

1. The presence of the duct structure can effectively reduce the influence of the upper propeller flow around and the tip vortices of the lower propeller on the effective lift area of the lower propeller while also reducing the dissipation of tip vortices and providing additional lift;

2. For the five influencing factors on the overall aerodynamic characteristics of the UAV, changing parameter settings within a certain range has corresponding effects on the four research targets. Within the range of the set operating conditions, these three research targets fluctuated with their respective analysis patterns, affecting the total lift. However, factors with higher sensitivity showed a faster increase in data. Therefore, combining this with Conclusion 3, making slight adjustments to the corresponding structural parameters will have a positive impact on the UAV.

3. In order to systematically study the influence of five factors on four research objects, the orthogonal experimental method was used to rank the sensitivity of the five factors. While ensuring overall strength, increasing the propeller spacing can result in a faster increase in lift for the upper propeller of the UAVs. Decreasing the distance between the propeller and the top surface of the duct can lead to a faster increase in lift for the lower propeller of the UAVs. Increasing the chord length of the duct cross-section can accelerate the lift of the duct structure and the overall lift of the UAVs.

**Author Contributions:** Conceptualization, H.X., L.Z., M.W. and Z.W.; Methodology, H.X., L.Z. and M.W.; Validation, H.X., L.Z., M.W., K.L. and H.Z.; Investigation, H.X., L.Z., M.W. and H.Z.; Resources, L.Z. and K.L.; Data curation, H.X., L.Z., M.W. and K.L.; Writing—original draft preparation, H.X. and L.Z.; Writing—review and editing, L.Z., K.L. and Z.W.; Software, H.X. and L.Z.; Supervision, L.Z. and Z.W.; Funding acquisition, H.X. and H.Z. All authors have read and agreed to the published version of the manuscript.

**Funding:** This research was funded by the Jiangsu Provincial Graduate Research and Practical Innovation Program for 2022, grant number SJCX22_0113.

**Data Availability Statement:** Data are contained within the article.

**Acknowledgments:** I would like to express my gratitude to the professor and colleagues for their assistance and encouragement regarding my research project.

**Conflicts of Interest:** The authors declare no conflict of interest.

## Nomenclature

| | |
|---|---|
| UAVs | Unmanned Aerial Vehicles |
| PIV | Particle Image Velocimetry |
| CFD | Computational Fluid Dynamics |
| RPM | Revolutions Per Minute |

## Appendix A

**Table A1.** Orthogonal experimental design for operating conditions.

| Number | D | d | S | $\alpha$ | L | None | Fan1 | Fan2 | Duct | Sum |
|--------|---|---|---|----------|---|------|------|------|------|-----|
| 1 | 3 | 1 | 1/6 L | −2 | 90 | 1 | 4.4144 | 1.8737 | 0.9252 | 7.2134 |
| 2 | 3 | 1.5 | 1/4 L | −1 | 95 | 2 | 4.1912 | 1.8000 | 1.6602 | 7.6514 |
| 3 | 3 | 2 | 1/3 L | 0 | 100 | 3 | 4.0378 | 1.7012 | 2.0547 | 7.7936 |
| 4 | 3 | 2.5 | 1/2 L | 1 | 105 | 4 | 3.9790 | 1.7336 | 2.0842 | 7.7968 |
| 5 | 3 | 3 | 7/12 L | 2 | 110 | 5 | 3.9176 | 1.6838 | 2.3012 | 7.9027 |
| 6 | 3.5 | 1 | 1/4 L | 0 | 105 | 5 | 4.0895 | 1.6619 | 2.6138 | 8.3652 |
| 7 | 3.5 | 1.5 | 1/3 L | 1 | 110 | 1 | 3.8821 | 1.6137 | 3.3467 | 8.8425 |
| 8 | 3.5 | 2 | 1/2 L | 2 | 90 | 2 | 3.9772 | 1.8356 | 1.4928 | 7.3056 |
| 9 | 3.5 | 2.5 | 7/12 L | −2 | 95 | 3 | 4.1147 | 1.9862 | 0.9169 | 7.0179 |
| 10 | 3.5 | 3 | 1/6 L | −1 | 100 | 4 | 4.4151 | 1.9290 | 0.9751 | 7.3192 |
| 11 | 4 | 1 | 1/3 L | 2 | 95 | 4 | 4.0277 | 1.8130 | 2.3175 | 8.1583 |
| 12 | 4 | 1.5 | 1/2 L | −2 | 100 | 5 | 3.9438 | 1.8593 | 2.3696 | 8.1726 |
| 13 | 4 | 2 | 7/12 L | −1 | 105 | 1 | 3.9511 | 1.8076 | 2.2084 | 7.9671 |
| 14 | 4 | 2.5 | 1/6 L | 0 | 110 | 2 | 4.3282 | 1.9092 | 1.3499 | 7.5872 |
| 15 | 4 | 3 | 1/4 L | 1 | 90 | 3 | 4.2663 | 1.8743 | 1.1535 | 7.2941 |
| 16 | 4.5 | 1 | 1/2 L | −1 | 110 | 3 | 3.8750 | 1.7062 | 3.6990 | 9.2802 |
| 17 | 4.5 | 1.5 | 7/12 L | 0 | 90 | 4 | 4.0123 | 2.0136 | 1.2483 | 7.2742 |
| 18 | 4.5 | 2 | 1/6 L | 1 | 95 | 5 | 4.4054 | 2.0392 | 1.0785 | 7.5231 |
| 19 | 4.5 | 2.5 | 1/4 L | 2 | 100 | 1 | 4.1697 | 1.9115 | 1.7948 | 7.8760 |
| 20 | 4.5 | 3 | 1/3 L | −2 | 105 | 2 | 4.0989 | 1.8910 | 1.7588 | 7.7487 |
| 21 | 5 | 1 | 7/12 L | 1 | 100 | 2 | 3.9523 | 1.8237 | 2.6617 | 8.4377 |
| 22 | 5 | 1.5 | 1/6 L | 2 | 105 | 3 | 4.3178 | 1.9779 | 1.4412 | 7.7369 |
| 23 | 5 | 2 | 1/4 L | −2 | 110 | 4 | 4.0780 | 1.8862 | 2.3833 | 8.3475 |
| 24 | 5 | 2.5 | 1/3 L | −1 | 90 | 5 | 4.1678 | 2.1129 | 1.1703 | 7.4510 |
| 25 | 5 | 3 | 1/2 L | 0 | 95 | 1 | 4.0810 | 2.0346 | 1.2430 | 7.3586 |
| 14 | 4 | 2.5 | 1/6 L | 0 | 110 | 2 | 4.3282 | 1.9092 | 1.3499 | 7.5872 |

**Table A2.** Sensitivity analysis of five structural factors for upward blade lift.

| Factors | 1(A) | 2(B) | 3(C) | 4(D) | 5(E) |
|---------|------|------|------|------|------|
| $K_{1j}$ | 20.5399 | 20.3589 | 21.8808 | 20.6498 | 20.8379 |
| $K_{2j}$ | 20.4787 | 20.3472 | 20.7947 | 20.6001 | 20.8200 |
| $K_{3j}$ | 20.5170 | 20.4495 | 20.2143 | 20.5488 | 20.5186 |
| $K_{4j}$ | 20.5612 | 20.7593 | 19.8560 | 20.4851 | 20.4363 |
| $K_{5j}$ | 20.5970 | 20.7790 | 19.9480 | 20.4100 | 20.0809 |
| $\overline{K_{1j}}$ | 4.1080 | 4.0718 | 4.3762 | 4.1300 | 4.1676 |
| $\overline{K_{2j}}$ | 4.0957 | 4.0694 | 4.1589 | 4.1200 | 4.1640 |
| $\overline{K_{3j}}$ | 4.1034 | 4.0899 | 4.0429 | 4.1098 | 4.1037 |
| $\overline{K_{4j}}$ | 4.1122 | 4.1519 | 3.9712 | 4.0970 | 4.0873 |
| $\overline{K_{5j}}$ | 4.1194 | 4.1558 | 3.9896 | 4.0820 | 4.0162 |
| $R_j$ | 0.0237 | 0.0864 | 0.4050 | 0.0480 | 0.1514 |
| $\sigma_j$ | 0.0089 | 0.0428 | 0.1669 | 0.0189 | 0.0624 |

**Table A3.** Sensitivity analysis of five structural factors for upward blade lift.

| Factors | 1(A) | 2(B) | 3(C) | 4(D) | 5(E) |
|---------|------|------|------|------|------|
| $K_{1j}$ | 8.7924 | 8.8785 | 9.7290 | 9.4965 | 9.7102 |
| $K_{2j}$ | 9.0264 | 9.2645 | 9.1339 | 9.3558 | 9.6731 |
| $K_{3j}$ | 9.2634 | 9.2698 | 9.1318 | 9.3205 | 9.2246 |
| $K_{4j}$ | 9.5615 | 9.6534 | 9.1693 | 9.0845 | 9.0720 |
| $K_{5j}$ | 9.8353 | 9.4127 | 9.3150 | 9.2218 | 8.7991 |
| $\overline{K_{1j}}$ | 1.7585 | 1.7757 | 1.9458 | 1.8993 | 1.9420 |
| $\overline{K_{2j}}$ | 1.8053 | 1.8529 | 1.8268 | 1.8712 | 1.9346 |
| $\overline{K_{3j}}$ | 1.8527 | 1.8540 | 1.8264 | 1.8641 | 1.8449 |
| $\overline{K_{4j}}$ | 1.9123 | 1.9307 | 1.8339 | 1.8169 | 1.8144 |
| $\overline{K_{5j}}$ | 1.9671 | 1.8825 | 1.8630 | 1.8444 | 1.7598 |
| $R_j$ | 0.2086 | 0.1550 | 0.1194 | 0.0824 | 0.1822 |
| $\sigma_j$ | 0.0830 | 0.0563 | 0.0507 | 0.0308 | 0.0785 |

**Table A4.** Sensitivity analysis of five structural factors for upward blade lift.

| Factors | 1(A) | 2(B) | 3(C) | 4(D) | 5(E) |
|---|---|---|---|---|---|
| $K_{1j}$ | 9.0256 | 12.21723 | 5.770018 | 8.353829 | 5.990154 |
| $K_{2j}$ | 9.3453 | 10.06599 | 9.605605 | 9.712941 | 7.216185 |
| $K_{3j}$ | 9.3989 | 9.21771 | 10.64799 | 8.509629 | 9.855815 |
| $K_{4j}$ | 9.5794 | 7.316109 | 10.88853 | 10.32459 | 10.10639 |
| $K_{5j}$ | 8.8995 | 7.431605 | 9.336496 | 9.347654 | 13.0801 |
| $\overline{K_{1j}}$ | 1.8051 | 2.4434 | 1.1540 | 1.6708 | 1.1980 |
| $\overline{K_{2j}}$ | 1.8691 | 2.0132 | 1.9211 | 1.9426 | 1.4432 |
| $\overline{K_{3j}}$ | 1.8798 | 1.8435 | 2.1296 | 1.7019 | 1.9712 |
| $\overline{K_{4j}}$ | 1.9159 | 1.4632 | 2.1777 | 2.0649 | 2.0213 |
| $\overline{K_{5j}}$ | 1.7799 | 1.4863 | 1.8673 | 1.8695 | 2.6160 |
| $R_j$ | 0.1360 | 0.9802 | 1.0237 | 0.3942 | 1.4180 |
| $\sigma_j$ | 0.0559 | 0.4064 | 0.4109 | 0.1652 | 0.5526 |

**Table A5.** Sensitivity analysis of five structural factors for upward blade lift.

| Factors | 1(A) | 2(B) | 3(C) | 4(D) | 5(E) |
|---|---|---|---|---|---|
| $K_{1j}$ | 38.3579 | 41.4547 | 37.3798 | 38.5001 | 36.5383 |
| $K_{2j}$ | 38.8504 | 39.6777 | 39.5342 | 39.6689 | 37.7093 |
| $K_{3j}$ | 39.1793 | 38.9370 | 39.9941 | 38.3789 | 39.5990 |
| $K_{4j}$ | 39.7022 | 37.7289 | 39.9138 | 39.8942 | 39.6147 |
| $K_{5j}$ | 39.3318 | 37.6233 | 38.5995 | 38.9794 | 41.9602 |
| $\overline{K_{1j}}$ | 7.6716 | 8.2909 | 7.4760 | 7.7000 | 7.3077 |
| $\overline{K_{2j}}$ | 7.7701 | 7.9355 | 7.9068 | 7.9338 | 7.5419 |
| $\overline{K_{3j}}$ | 7.8359 | 7.7874 | 7.9988 | 7.6758 | 7.9198 |
| $\overline{K_{4j}}$ | 7.9404 | 7.5458 | 7.9828 | 7.9788 | 7.9229 |
| $\overline{K_{5j}}$ | 7.8664 | 7.5247 | 7.7199 | 7.7959 | 8.3920 |
| $R_j$ | 0.2689 | 0.7663 | 0.5229 | 0.3031 | 1.0844 |
| $\sigma_j$ | 0.1017 | 0.3157 | 0.2204 | 0.1359 | 0.4146 |

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
