# Peer review of "Analysis of the Impact of Structural Parameter Changes on the Overall Aerodynamic Characteristics of Ducted UAVs"

_drones, doi:10.3390/drones7120702_

Round 1

Reviewer 1 Report

Comments and Suggestions for Authors

The manuscript presents an interesting and relevant study toward designing a better UAV. I recommend a minor revision. 

Author Response

Dear reviewer, thank you for pointing these out. Please refer to the WORD file below for specific details. Once again, thank you for your assistance and guidance.

Reviewer 2 Report

Comments and Suggestions for Authors

The manuscript investigated the effects of structural parameters such as co-axial twin propeller configuration, duct structure on the aerodynamic performance of ducted UAVs via CFD and experiment. The impact sensitivity analysis was conducted by the orthogonal test method. It is interesting and well organized. It could be accepted after a minor revision. 

1.     Fig. 3, the duct cross-sectional configuration may have a significant effect on the aerodynamic performance of the ducted UAV. Have you investigated this issue? 

2.     Fig. 7, to verify the influence of time step, how many cells are used in the computation? It is hard to be understood that smaller time steps result in larger deviation of thrusts.

3.     Line 214, section 3.3, the lift on the individual propeller and the coaxial dual-rotor without a duct were measured. Why didn't you also conduct the testing of the coaxial dual-rotor with a duct to obtain exact thrusts as the target numerical simulation?

4.     Fig. 17, the lift forces on the duct and the upper fan don't vary with rotor spacing. The lift on the lower fan increases with spacing increasing. A physical interpretation of the phenomena is suggested.

5.     Line 353, section 5. the influence of various structural parameters on the lift of a ducted UAVs is investigated via an orthogonal experimental design. This reviewer believes that the optimized structural parameters could be found. It is recommended to describe those parameters and the overall aerodynamic performance of the UAVs. 

Comments on the Quality of English Language

The English writing is generally good. However, in section 4.2, several phrases 'in order to' were used. It is better to replace them with the other words.

Author Response

(The authors gave the same response as above.)

Reviewer 3 Report

Comments and Suggestions for Authors

The authors presented a numerical study on the effect of Structural Parameter Changes on the Overall Aerodynamic Characteristics of Ducted UAVs.

The paper is generally well prepared, has good scientific soundness and can be accepted for publication after addressing the following points:

The introduction is relatively short and may be extended by adding recently published papers related to the subject.

The novelty of the paper is to be clearly stated.

The solved governing equations are to be presented.

The used turbulence model is to be justified.

The boundary conditions are to be expressed mathematically.

More details on the experimental setup, measurement techniques and data acquisition system are to be provided.

Details on the performances of the used computer and computational time are to be provided.

A figure presenting the 3D mesh is to be added.

A figure presenting the 3D flow structure (streamlines) is to be added.

The paper is to be checked for misprints and grammatical errors.

Comments on the Quality of English Language

The paper is to be checked for misprints and grammatical errors.

Author Response

(The authors gave the same response as above.)

Reviewer 4 Report

Comments and Suggestions for Authors

This paper conducted a CFD simulation of Analysis of the Impact of Structural Parameter Changes on the Overall Aerodynamic Characteristics of Ducted UAVs. Major revisions should be done before it can be considered for publication.

1. The number of literature cited by the authors is too small, and there have been very few new literature in recent years, as well as there are too many Chinese literature and master's and doctoral theses. So, the Introduction should be written.

2. Figure 9 agrees too well. Please provide detailed raw test data (preferably a complete video) to prove the authenticity of the data in the figure.

3. Figure 14, Figure 15, Figure 16, the maximum value cannot be seen in which areas in the figures.

Comments on the Quality of English Language

The English of the whole paper should be revised by native speakers.

Author Response

(The authors gave the same response as above.)

Round 2

Reviewer 4 Report

Comments and Suggestions for Authors

No more comments.